# The Effects of Lignin on the Thermal and Morphological Properties and Damage Mechanisms after UV Irradiation of Polypropylene Biocomposites Reinforced with Flax and Pine Fibres: Acoustic Emission Analysis

**DOI:** 10.3390/ma17112474

**Published:** 2024-05-21

**Authors:** Zouheyr Belouadah, Khaled Nasri, Lotfi Toubal

**Affiliations:** 1Laboratoire des Sciences et Techniques de l’Environnement, Ecole Nationale Polytechnique, 10 Avenue des Freres Oudek, BP 182, El-Harrach, Alger 16200, Algeria; zouheyr.belouadah@g.enp.edu.dz; 2Mechanical Engineering Department, Innovations Institute in Ecomaterials, Ecoproducts and Ecoenergy (I2E3), Université du Quebec à Trois-Rivières, 3351 Boul. des Forges, Trois Rivières, QC G9A 5H7, Canada; nasri.khaled@uqtr.ca

**Keywords:** biocomposites, UV ageing, acoustic emission, damage mechanisms, environmental degradation, thermogravimetric analysis

## Abstract

This study investigates the impact of lignin on the durability and performance of polypropylene-based biocomposites (PP–flax and PP–pine) under environmental stresses such as UV radiation and moisture. The findings indicate that pine fibres, with their higher lignin content, are significantly more resistant to thermal degradation than flax fibres. Differential scanning calorimetry (DSC) showed that lignin influences crystallinity and melting temperatures across the composites, with variations corresponding to fibre type. Acoustic emissions analysis revealed that increasing the lignin content in pine fibres effectively reduces surface microcracks under UV exposure. Overall, these results underscore the importance of fibre composition in improving the performance and longevity of biocomposites, making them better suited for durable construction applications.

## 1. Introduction

The integration of environmentally friendly materials in product design has contributed to the growth of the natural fibres and biocomposites market, with revenues expected to exceed USD 61.04 billion by 2030 [1]. The high reinforcing capability of these fibres and their small environmental footprint compared to those of synthetic fibres have led to their increasing use in various industrial sectors, such as the automotive and construction sectors. For instance, in the construction sector, particularly in North America, biocomposites, also known as wood–plastic composites (WPCs) or natural fibre composites (NFCs), are widely used in the manufacturing of door and window frames, solid and hollow boards, and laminated flooring [2]. However, structures exposed to the outdoors undergo both ultraviolet (UV) and moisture-induced damage, negatively affecting their mechanical properties [3,4]. The mechanical behaviour of natural short-fibre composites has been the subject of several studies [5,6,7]. Most of these studies examine the influence of fibre content, manufacturing processes, and the choice of coupling agent on the mechanical properties and material damage mechanisms. Regarding ageing, previous studies generally relied on qualitative assessments (such as investigating changes in colour, for example), without establishing a link between the influence of UV radiation on mechanical properties and the chemical composition of the fibres. Understanding how the variable lignin content affects the mechanical degradation and damage mechanisms of wood–plastic composites (WPCs) under UV ageing conditions is crucial for assessing their performance, but this has not been thoroughly studied. This allows us to not only better understand how different types of fibres interact with the matrix but also examine the evolution of these interactions over time. Additionally, in this work, the emphasis on the matrix–fibre interaction is particularly relevant in the context of practical applications, where the long-term performance of the composite is a major concern. By studying the degradation of the composite in the presence of different types of fibres, informed decisions can be made regarding the selection and design of materials for specific applications.

The natural ageing process, driven by unpredictable variations in temperature, humidity, and other environmental factors, leads to complex and unpredictable material degradation. In contrast, environmental chamber testing exposes materials to controlled conditions, including elevated temperatures, light intensities beyond those typically encountered outdoors, and shorter wavelengths of light. While accelerated ageing tests do not perfectly replicate natural ageing, they provide a significant advantage by simulating the ageing of biocomposites in just a few weeks, a process that would take months or even years outdoors [8].

Biocomposites with short fibres, typically made from conventional thermoplastics like polypropylene (PP) and polyethylene (PE), are amalgamated with short natural fibres ranging from 0.1 to 2 mm. The interaction between the type of natural fibres and the matrix, particularly the hydrophilic nature of these fibres and their limited UV resistance, leads to fibre–matrix decohesion, impairing the transfer of interfacial stresses [4,5,9,10,11,12,13].

This investigation specifically focuses on flax and pine fibres. Flax fibres, which are rich in cellulose, contrast sharply with pine fibres, which are notable for their high lignin content (28% compared with approximately 3% for flax [6,7]). Cellulose and hemicellulose contribute to the yield strength and rigidity of biocomposites, while the hydrophobic nature of lignin serves as a binding agent within the fibres [6,7,14,15].

The primary objective of this study is to explore the specific role of lignin in the performance of biocomposites under environmental stresses such as UV and moisture exposure. This focus is particularly relevant for assessing how the chemical composition affects the morphological characteristics, thermal degradation, and UV-induced damage in polypropylene/flax (PP–flax) and polypropylene/pine (PP–pine) fibre composites. Understanding these interactions is essential for advancing the development of durable and resilient materials suited for various industrial applications.

One can measure the initial properties of biocomposites, but predicting their evolution over time remains challenging. Exposing biocomposites to UV radiation breaks covalent bonds, resulting in discoloration, yellowing, surface roughness, weight loss, the deterioration of mechanical properties, and cracks [6,7,12,13,16]. For fibres, the ageing process begins when UV radiation is absorbed by the lignin structure, forming chromophore groups, such as carboxylic acids, quinones, and hydroperoxyl radicals, responsible for yellowing and discoloration [16]. The photodegradation of polymers induces surface oxidation, chain scission, and, consequently, the breaking of bonding molecules. This process of degradation in polymer matrices leads to the formation of superficial cracks, significantly decreasing the tensile strength of biocomposites. Comprehensive literature reviews on the degradation mechanisms of biocomposites under UV exposure are presented in [5,6,7].

In this study, biocomposites were manufactured from pellets of a polypropylene biocomposite reinforced with 30% short flax fibres (PP–flax) or with 30% short pine fibres (PP–pine) obtained from Rhetech, Inc. (Whitmore Lake, MI, USA). Due to the lack of detailed characteristics regarding the constituents of composite pellets, particularly natural fibres, a thorough understanding of the individual properties of these components is essential for analysing final product behaviour. To address this, the separation of the matrix (PP) from the natural fibres (flax and pine) was performed. This process enabled the assessment and, more importantly, comparison of the mechanical, physical, morphological, and thermal properties of the studied materials at the scale of fibres and the composite itself. Furthermore, acoustic emission (AE) testing was utilized during tensile tests to study the progression of various damage mechanisms that ultimately lead to specimen failure. This technique offers advantages in terms of sensitivity to different types of damage in composites [17]. AE parameters such as amplitude, duration, and energy, were selected as evaluation indices to correlate with the material’s mechanical behaviour. Thus, damages related to UV/humidity ageing were documented, and the influence of the chemical and morphological properties of these fibres on the damage threshold was better assessed. According to studies conducted thus far, the current study highlights the response of both types of fibres to the effects of UV degradation in these biocomposites.

## 2. Experiments

### 2.1. Materials, Specimens, and Tensile Testing

Composite pellets of PP reinforced with 30% short pine fibres (PP–pine) (Figure 1a) or 30% short flax fibres (PP–flax) (Figure 1b) were used to shape the tensile specimens, as shown in Figure 1. Considering the direct impact of the plant fibre chemical composition on the UV effect on composite durability and to facilitate better identification of the composite damage mechanisms, pellet composition was analysed by separating the different components and subsequently examining each constituent individually. The process of fibre–matrix separation is illustrated in Figure 2.

Approximately 20 g of composite pellets were placed in an Erlenmeyer flask containing 150 mL of xylene solution. The mixture was stirred at 1500 rpm and heated to approximately 140 °C until the pellets had completely dissolved. The fibres were then separated from the matrix using a sieve. This procedure was repeated several times until the complete removal of PP from the surface of the fibres. The obtained pine and flax fibres were placed in an oven at 100 °C for several hours until the fibres were completely dry. The separated polypropylene was air-dried.

Utilizing the 100-TON HAITIAN ZHAFIR ZTR 900 press (Zhafir Zeres series ZE900/210, Haitian, Inc., Ningbo, China), injection moulding was performed specifically for the production of tensile samples in accordance with ASTM D-638 [18]. To mitigate the occurrence of micro-voids and porosity in the samples post-injection, the biocomposite pellets were subjected to a pre-injection drying phase at 80 °C for 2 h. The injection process was conducted at 200 °C, with a pressure of 15,000 PSI, an injection temperature of 180 °C, and an injection time of 20 s, all of which were recommended by the manufacturer and meticulously followed to ensure the accurate and efficient injection of the samples.

Following sample injection, the samples were subjected to optimal cooling at a temperature of 60 °C, as specified by the manufacturer, to facilitate the proper solidification of the material. Additionally, an appropriate waiting time after moulding was observed, as recommended by the manufacturer, to allow for the complete solidification of the material before removing the samples from the mould.

### 2.2. Characterization Methods

#### 2.2.1. True Density Measurement

To determine the true density of PP–pine and PP–flax composite pellets and their constituents—pure PP, flax fibres, and pine fibres—a gas pycnometer was employed. Approximately 3 g of a powder sample was analysed ten times using a Micromeritics AccuPyc II 1340 (Micromeritics Japan, G.K, Kashiwa, Japan) at 24 °C under helium gas (He) conditions.

#### 2.2.2. Thermal Assessment

The temperature profiles for the thermogravimetric analysis (TGA) and differential scanning calorimetry (DSC) were determined from ambient conditions to 600 °C in a nitrogen environment with a heating rate of 10 °C/min using TA Instruments apparatus (TA Instruments, Guyancourt, France).

#### 2.2.3. ATR–FTIR Analysis

The chemical composition of the PP–pine and PP–flax composite pellets and their constituents (PP, flax fibres, and pine fibres) were examined using the attenuated total reflectance–Fourier transform infrared spectroscopy (ATR–FTIR) technique. This method is a powerful tool for qualitative analysis. ATR–FTIR measurements were conducted with the IRTracer-100_NIS-PC-Instrument (SHIMADZU, Marne la Vallée, France) at room temperature for wavenumbers ranging from 4000 to 400 cm^−1^ and a spectral resolution of 4 cm^−1^.

#### 2.2.4. Morphological Analysis

The aspect ratio of the fibres, representing the relationship between the length (*L*) and diameter (*D*) of the fibres incorporated in the composite pellets (*L*/*D*), was evaluated to assess this essential parameter that influences the mechanical and physical properties of the final material. For this evaluation, the morphology of the short fibres was thoroughly examined using an optical microscope. The samples were carefully prepared and observed at various levels of magnification. This morphological analysis yielded a detailed view of the fibre characteristics, including their size, shape, and uniformity.

### 2.3. Artificial Weathering

The industry demands faster accelerated weathering test results while ensuring a strong correlation with real-time exposure outcomes, and these demands persist in industry laboratory simulations. Here, the ageing conditions were established according to ASTM G154-23 [19], a standard practice for the artificial UV ageing of non-metallic materials. It is understood that the PP matrix degrades under the influence of UV rays, justifying its selection as a reference in the comparative study of biocomposites. However, the effect of UV ageing on virgin PP was not examined for two reasons. Initially, access to the PP used by Rhetech to manufacture the pellets desired for this research was unavailable. Secondly, pure PP exhibits significant toughness [11], and the primary objective of this research was to analyse the impact of UV rays on the performance of biocomposites, with a particular focus on how the chemical composition of natural fibres influences their degradation. The ASTM G154 standard provides general guidelines for selecting an appropriate exposure duration. It suggests starting with a relatively short exposure and gradually increasing the duration until an adequate level of degradation is achieved. Generally, an exposure of 1000 to 2000 h under ASTM G154 test conditions can represent several years of real-world exposure. However, an accurate estimation of the exposure time would depend on several factors, such as climate, geographical location, local weather conditions, etc. The choice of 1400 h is often made, and it falls within the range of durations specified in the ASTM G154 standard. Two environmental conditions were considered in this study:UV without humidity: The samples were subjected to UV ageing using UVA-340 fluorescent lamps (irradiance at a wavelength of 340 nm) in a QUV/SE ageing apparatus (Q-Lab Co., Westlake, OH, USA). Ageing lasted for two months, with 8 h of UV exposure at an irradiance of 1.55 W/m^2^ at 60 °C per day.UV with humidity: The samples, consistent with those in Condition 1, underwent two months of ageing but included 4 h of water condensation at 50 °C following each UV exposure session at 60 °C.

### 2.4. Acoustic Emission

Acoustic emission (AE) measurements were conducted using two resonant MICRO-80 sensors operating within a 100–1000 kHz frequency range, utilizing a system provided by the Physical Acoustics Corporation (MISTRAS, Columbia, SC, USA). Prior to each test, three pencil lead break tests were performed to calibrate the source location of AE events and to assess the coupling state between the specimen and the sensors. The recorded AE acoustic signals were analysed, focusing on parameters such as amplitude, count, duration, and frequency, as referenced in [11,15,17]. By employing the K-means clustering algorithm, each set of acoustic events was classified into a specific damage mechanism.

## 3. Results and Discussion

### 3.1. Morphological Analysis

The length-to-diameter ratio (*L*/*D*) plays a pivotal role in the mechanical properties of composite materials composed of flax and pine fibres. An optimal *L*/*D* ratio enhances the load transfer among components, thereby reinforcing their mechanical attributes. Inadequate values can create weak points, compromising the overall strength of the material. Achieving a precise balance of *L*/*D* is essential for optimizing the mechanical performance of composites. Figure 3 displays microscopic images of the fibres, showing *L*/*D* values of 4.82 ± 2.76 for flax and 3.17 ± 1.43 for pine.

### 3.2. True Density Measurement

Density and porosity analysis provide significant insights into the physical characteristics and potential performance of various materials, as depicted in Figure 4. The plant fibres, despite their similar categorization, exhibit distinct densities of 1.51 g/cm³ for flax and 1.47 g/cm³ for pine, along with porosities of 34% and 32%, respectively. These variations may have been influenced by the differences in growth conditions and the compositional variances of the substances, such as those regarding hemicellulose, lignin, and impurities.

Compared to pure polypropylene PP, which has a density of 0.98 g/cm³ and a porosity of 2%, the composites display variations: PP–flax has a slightly decreased density of 0.96 g/cm³ and an increased porosity of 3%. This suggests potential post-fabrication changes and a higher length-to-diameter (*L/D*) ratio of the fibres. Conversely, PP–pine exhibits an increased density of 1.01 g/cm³ with a reduced porosity of 1.5%, indicative of more effective fibre integration and a lower *L/D* ratio for the fibres. These differences not only distinguish each material but also highlight the intricate effects of fibre type, fibre–matrix interactions, and fabrication processes on composites’ properties. This information provides valuable avenues for targeted material design and application-specific research. The densities of PP, pine fibre, and flax fibre align with those reported in previous studies [20,21,22].

### 3.3. Thermogravimetric Analysis

Thermal stability holds significance for natural fibres considered as potential reinforcements in composite materials. Manufacturing such materials often involves treatments at elevated temperatures, particularly when the composite matrix is a thermoplastic. The thermal behaviour of composites is significantly influenced by the chemical composition of the fibres.

Figure 5a,b illustrate the respective thermogravimetric (TG) curves and first derivatives (DTG) for the composite pellets and their constituents. These curves reveal three distinct stages of thermal change as the temperature increases. The first stage is specifically associated with flax and pine fibres. Up to 100 °C, the decreases in mass by 5.57% and 7% for the pine and flax fibres, respectively, are due to moisture loss, as reported in some works [23,24]. Subsequently, both fibre types maintain thermal stability until reaching 238 °C for flax fibre and 247 °C for pine fibre. This differentiation stems from the higher lignin content in pine fibres, thus retarding the thermal degradation of plant fibres. Manral et al. [25] highlighted that cellulose and hemicellulose degrade because of heat faster than lignin. After the TGA analysis, the char yield was found to be 25.99% for flax fibres and 14.19% for pine fibres, reflecting the differential impact of lignin content on char formation.

The second stage relates only to fibres and composite pellets and occurs at 238 °C to 390 °C for flax and pine fibres and 260 °C to 378 °C for PP–flax and PP–pine. This phase sees the significant decomposition of the fibre’s main components. In the case of pine fibre, a 72% mass loss indicates a delay in decomposition compared to flax fibre, which experiences a 54% mass loss, owing to the higher lignin content in pine fibres. Burhenne et al. [26] observed that the TG curve of spruce wood with bark shifts approximately 253 °C higher than the TG curves of straw and rape straw. This implies that breaking down woody biomass demands increased activation energy because of its elevated lignin content and distinct type when contrasted with straw. The same observation was reported by Kristanto et al. [27]. Of note, the thermal behaviour of fibres also impacts that of composite pellets. Slower thermal degradation is observed for PP–pine pellets, resulting in a 28% mass loss, while PP–flax pellets lose 13% mass. Several studies have demonstrated enhanced thermal stability in various materials following the incorporation of lignin. The impact of lignin content significantly enhances the thermal stability of materials by increasing their initial decomposition temperatures and oxidation induction times. This improvement is attributed to the phenylpropanoid units, whose aromatic structures exhibit considerable stability primarily due to the overlapping of p-orbitals, allowing for the complete localization of π electrons. This molecular arrangement enhances the material’s heat resistance prior to degradation [28,29,30]. In the third stage, which extends to 600 °C, additional losses of 12% for flax fibre and 8% for pine fibre are recorded.

### 3.4. Differential Scanning Calorimetry Analysis

Figure 6 illustrates the DSC curves for flax, pine, PP, PP–pine, and PP–flax, and the analysis results are presented in Table 1. An endothermic peak, observed at temperatures below 105 °C for all samples, is attributed to the evaporation of water present in natural fibres and PP. This peak aligns with the corresponding peak observed in the TG curves, indicating its direct relation to the water content of the samples, which has been further evaluated to understand the effects of post-processing on moisture retention. Notably, the water content tends to be higher in natural fibres than in PP and composites (PP–pine, PP–flax), reflecting variations influenced by the type of fibre, processing, storage conditions, and humidity levels. The three samples containing PP exhibited a peak at 162–163 °C, corresponding to the melting point of the PP. The calculated enthalpy around this peak is notably higher for PP–flax compared to pure PP and PP–pine. The enthalpy of this peak is directly proportional to the fraction of crystallinity of the material, a value calculated using the following equation:(1)$$F_{c}=∆H_{m}/\left[{∆H}_{m}^{0}\left(1-x\right)\right]$$
where *F_c_* is the crystal fraction, Δ*H_m_* is the enthalpy of fusion measured at the peak melting temperature of PP, and ${∆H}_{m}^{0}$ signifies the enthalpy of fusion of the crystal phase of PP at 100% crystallinity. The latter was determined using the value from [31], specifically 209 J/g.

The degree of crystallinity of PP depends on factors such as cooling rate, the degree of branching, molecular orientation, and the presence of additives or fillers [32]. In this study, it was higher for PP–flax (12%) than for pure PP (9%) and PP–pine (6%). The increase in crystallinity for flax fibre-reinforced pellets resulted from the positive effect of nucleation of flax fibre and the high content of crystalline components in flax fibres such as cellulose [33,34]. In contrast, the decrease in crystallinity in the case of PP–pine was due to the dilution effect of pine fibre [35] and the presence of a high percentage of amorphous components such as lignin.

Malkapuram et al. [35,36] assessed the thermal behaviour of pine needle fibre-reinforced PP composites. They observed a decrease in the degree of crystallinity with increased fibre content. The degree of crystallinity of PP and composite pellets had implications for their mechanical, thermal, and electrical properties [37]. In our study, both natural fibre samples exhibited a peak at 249–250 °C, signifying the initiation of thermal degradation in main components such as cellulose, hemicellulose, and lignin, as evidenced in the TGA analysis. The calculated enthalpy values around this peak were 63 J/g for flax fibre and 41 J/g for pine fibre. The enthalpy of this peak is directly proportional to the type and chemical composition of the natural fibres. The chemical composition of natural fibres varies based on factors including fibre type, plant part, processing, and ageing. The chemical composition of flax fibre was primarily dominated by cellulose (68.7–75.5%), followed by hemicellulose (12.2–14%) and lignin (2.1–4.7%). In contrast, the chemical composition of pine fibre comprised 42% cellulose, 29% hemicellulose, and 28% lignin [12,15]. The peak observed at 460–464 °C corresponded to the thermal degradation of PP. The calculated enthalpy values at this peak were 662 J/g for PP, 560 J/g for PP–flax, and 427 J/g for PP–pine.

The TGA results confirm the four heat flow peaks identified in the DSC analysis, corresponding to the stages of thermal degradation observed: water evaporation, PP melting, degradation of the main components of natural fibres, and PP degradation. The PP melting phenomenon is more distinctly evident in the DSC analysis. Both TGA and DSC outcomes provide insights into the thermal properties of flax fibres, pine fibres, and composite materials, encompassing parameters such as water content, crystallinity degree, thermal stability, melting temperature, and decomposition temperature.

### 3.5. ATR–FTIR Analysis

The ATR–FTIR test curves of Figure 7 show absorption bands characteristic of the different functional groups in the five samples (PP, PP–pine, PP–flax, pine fibres, and flax fibres). The absorption bands are proportional to the concentration of the functional groups in the samples. The flax and pine fibres show similar absorption bands, corresponding to the functional groups of cellulose, hemicellulose, and lignin, the main components of natural fibres. The most critical absorption bands are those at 3340 cm^−1^ (H–O stretching vibrations), 2920 cm^−1^ (C–H stretching vibrations), 1734 cm^−1^ (C=O stretching vibrations), 1593 cm^−1^ (C=C aromatic stretch), 1234 cm^−1^ (acetyl group C–O bond stretching vibrations), 1107 cm^−1^ (C–O stretching vibrations of the ester group), 1049 cm^−1^ (C–O–C stretch), and 796 cm^−1^ (C–H deformation) [24,38,39].

The pine fibre shows higher-intensity absorption bands at 1734 cm^−1^ and 1234 cm^−1^ than the flax fibre, indicating a higher hemicellulose and lignin content in the pine fibre. Hemicellulose and lignin are the most amorphous components, and lignin is the most thermally stable component of natural fibres. The PP and the composites reinforced with flax and pine fibres show similar absorption bands corresponding to the functional groups of PP, which is the polymer matrix of the composites. The most significant absorption bands are those at 2949 cm^−1^ (CH_3_ asymmetric stretching), 2916 cm^−1^ (symmetric stretching of CH_2_), 2848 cm^−1^ (CH_2_ asymmetric stretching), 1462 cm^−1^ (CH_3_ symmetrical bending), 1375 cm^−1^ (CH_3_ symmetric deformation vibration), 1166 cm^−1^ (C–H rocking), 997 cm^−1^ (CH_3_ asymmetric rocking vibration), 972 cm^−1^ (CH_3_ asymmetric rocking and the CC asymmetric stretching vibrations), 840 cm^−1^ (CH_3_ Rocking), and 719 cm^−1^ (C–C stretching) [40,41,42].

The PP shows higher-intensity absorption bands at 2949 cm^−1^, 2916 cm^−1^, 2848 cm^−1^, 1462 cm^−1^, 1375 cm^−1^, 1166 cm^−1^, 997 cm^−1^, 972 cm^−1^, 840 cm^−1^, and 719 cm^−1^ than the composites reinforced with flax and pine fibres, indicating a higher content of PP compared to the composite pellets. PP is the most thermoplastic and fusible component of the composites. Additionally, the peaks observed in the curves of the pine and flax fibres do not appear in the case of PP, PP–pine, and PP–flax. Therefore, a good coverage of the flax and pine fibre by the PP matrix is present. The same observation was recorded by Alam et al. [43].

### 3.6. Acoustic Emission Analysis

Acoustic emission analysis serves as a non-destructive method to monitor degradation processes in real-time. It detects and analyses acoustic signals emitted by samples under mechanical stress or degradation conditions. In this study, the technique was used to detec microcrack propagation, particularly microcracks resulting from UV-induced damage. It facilitated the identification of how these microcracks contribute to the total material fracture post-tensile testing [13]. A recent study by Ferdinánd et al. [44] highlighted the critical role of fibre–matrix interactions and the mechanical integrity of fibre-reinforced composites under dynamic stresses.

Figure 8 and Figure 9 illustrate the localization and distribution of AE signals or “hits” recorded during the tests, analysed based on three parameters: burst amplitude, counts, and duration. The results are presented for non-aged specimens (Figure 8a and Figure 9a), specimens aged under dry conditions (Figure 8b and Figure 9b), and specimens aged under humidity (Figure 8c and Figure 9c) concerning the PP–flax and PP–pine samples, respectively. The amplitude range of AE signals associated with matrix cracking, matrix–matrix friction, decohesion, and fibre–matrix friction varies from 40 to 50 dB, 50 to 60 dB, 55 to 75 dB, and 60 dB and above, respectively. Additionally, it is observed that AE events are concentrated in the areas where the tested specimens have ruptured.

Results indicate that both materials exhibit similar damage mechanisms, including matrix cracking, matrix–matrix friction, decohesion, and fibre–matrix friction. However, the percentages of these mechanisms vary between the two materials. Prior to ageing, PP–flax showed lower levels of decohesion compared to PP–pine at 12% and 21%, respectively. The predominant mechanism in both materials is matrix cracking, accounting for 56% in PP–flax and 64% in PP–pine. After ageing, a significant increase in the percentage of matrix cracking was observed in both materials, possibly due to the formation of superficial microcracks under UV exposure, as increased lignin in natural fibres enhances photo-oxidation reactions [12]. These superficial microcracks, which were generated at the PP matrix level, propagated during tensile tests, leading to an increase in the percentage of matrix cracking after ageing. Figure 8a–c reveal an 18% and 19% increase in matrix cracking for Conditions 1 and 2 for PP–flax, while PP–pine shows increases of 11% and 14%, respectively. Biocomposites with a higher lignin content, like those found in pine fibres, showed greater resistance to degradation and reduced microcrack formation compared to those with a lower lignin content. UV rays are largely absorbed by lignin, protecting the PP matrix from direct UV attack. Microcracks induced by UV propagate during tensile testing, primarily affecting the PP matrix or causing polymer chain scission. When both materials were exposed to UV rays or high temperatures, photo-oxidation reactions occurred in the lignin of natural fibres, with moisture accelerating these reactions. The rate of photo-oxidation reactions in biocomposites depends on the chemical composition of the natural fibres. The substantial presence of lignin in pine fibres can act as a UV absorber, attenuating damage and degradation in PP–pine composites compared to those containing flax fibres. Additionally, analysis of the acoustic emission data revealed significant differences in damage mechanisms between the two materials under UV exposure. It was observed that UV radiation promotes or exacerbates matrix-cracking mechanisms in both materials. Furthermore, the matrix cracking of outdoor biocomposite structures deteriorates considerably more in humid environments than in dry ones due to heightened chemical reactivity [7,11,13]. Tests were conducted three times, demonstrating high precision with minimal standard deviations in all measurements, ensuring the reliability of the data obtained. These observations underscore the impact of UV ageing on the damage mechanisms of biocomposites and emphasize the necessity of comprehending these phenomena for the advancement of durable and resilient materials for various industrial applications.

## 4. Conclusions

This study provides a comprehensive overview of the effect of the chemical composition of plant fibres on the UV ageing behaviour of biocomposites reinforced with flax and pine fibres. After analysing PP–flax and PP–pine fibre composites and their components in detail, several significant conclusions emerged. First, the chemical composition of plant fibres, particularly the lignin content, directly impacts the durability of composites against UV effects. Precisely identifying the composition of the biocomposite pellets enables a deeper understanding of their thermal behaviour and potential damage mechanisms. These insights are critical for developing durable materials.

Microscopic images of the fibres revealed distinct characteristics regarding the length-to-diameter ratio (*L*/*D)* and density, with different values for flax and pine. These differences are also reflected in the thermal properties of the composites, wherein the lignin content of pine fibres delays thermal degradation compared to flax fibres.

Thermogravimetric (TG) and differential scanning calorimetry (DSC) analyses have provided insights into the thermal properties of flax fibres, pine fibres, and composite materials. These analyses encompass parameters such as water content, crystallinity degree, thermal stability, melting temperature, and decomposition temperature. ATR–FTIR spectra identified functional groups present in natural fibres and polypropylene (PP), thus strengthening the understanding of the interaction between the composite phases.

Acoustic emission analysis (AE) monitored degradation processes by detecting and analysing signals emitted by samples under stress or degradation, aiding in understanding material fracture mechanisms. The results showed variations in signal localization and distribution, highlighting different damage mechanisms. Both materials exhibited similar damage mechanisms but with varying levels of resistance to degradation.

These findings underscore the importance of continuing research into the development of composites resistant to environmental conditions, considering the specific properties of natural fibres and their interaction with the polymer matrix.

## Figures and Tables

**Figure 1 materials-17-02474-f001:**
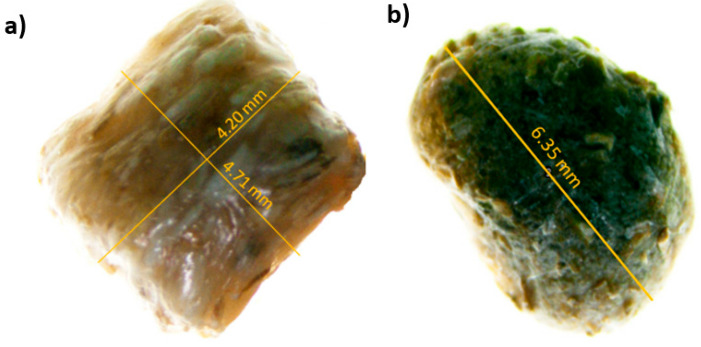
Composite pellets of PP reinforced with (**a**) 30% short pine fibres and (**b**) 30% short flax fibres.

**Figure 2 materials-17-02474-f002:**
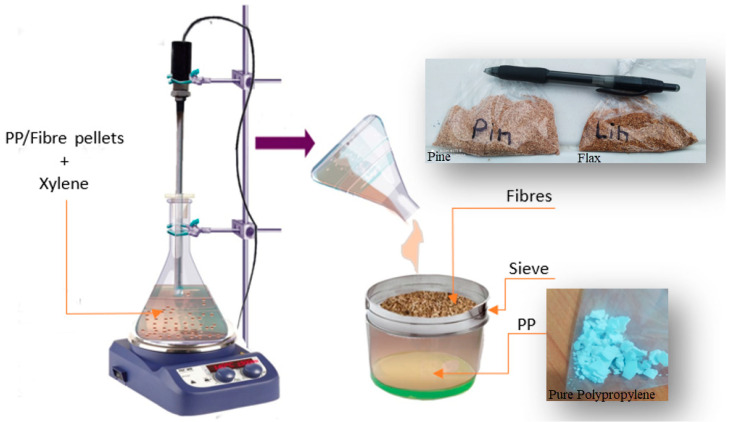
Fibre–matrix separation process.

**Figure 3 materials-17-02474-f003:**
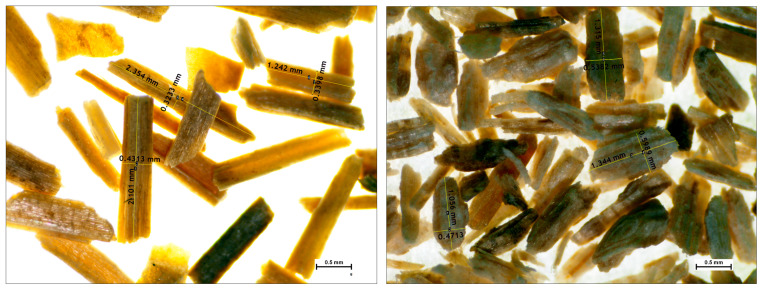
Microscopic micrographs of flax and pine fibres after separation from the matrix.

**Figure 4 materials-17-02474-f004:**
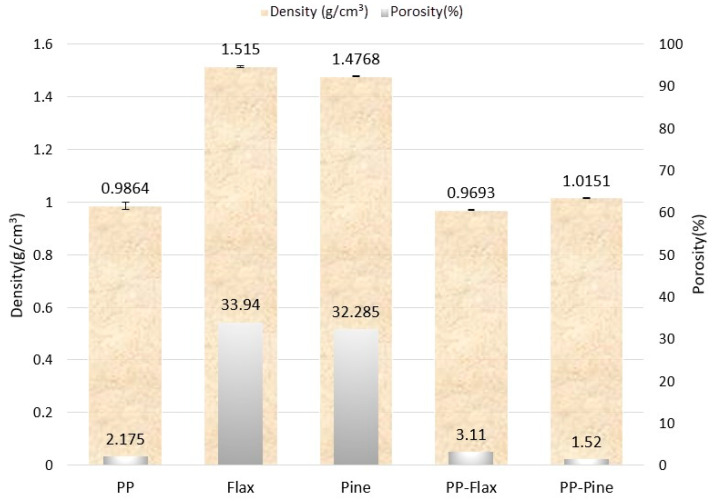
Comparison of density and porosity for composite pellets and their constituents.

**Figure 5 materials-17-02474-f005:**
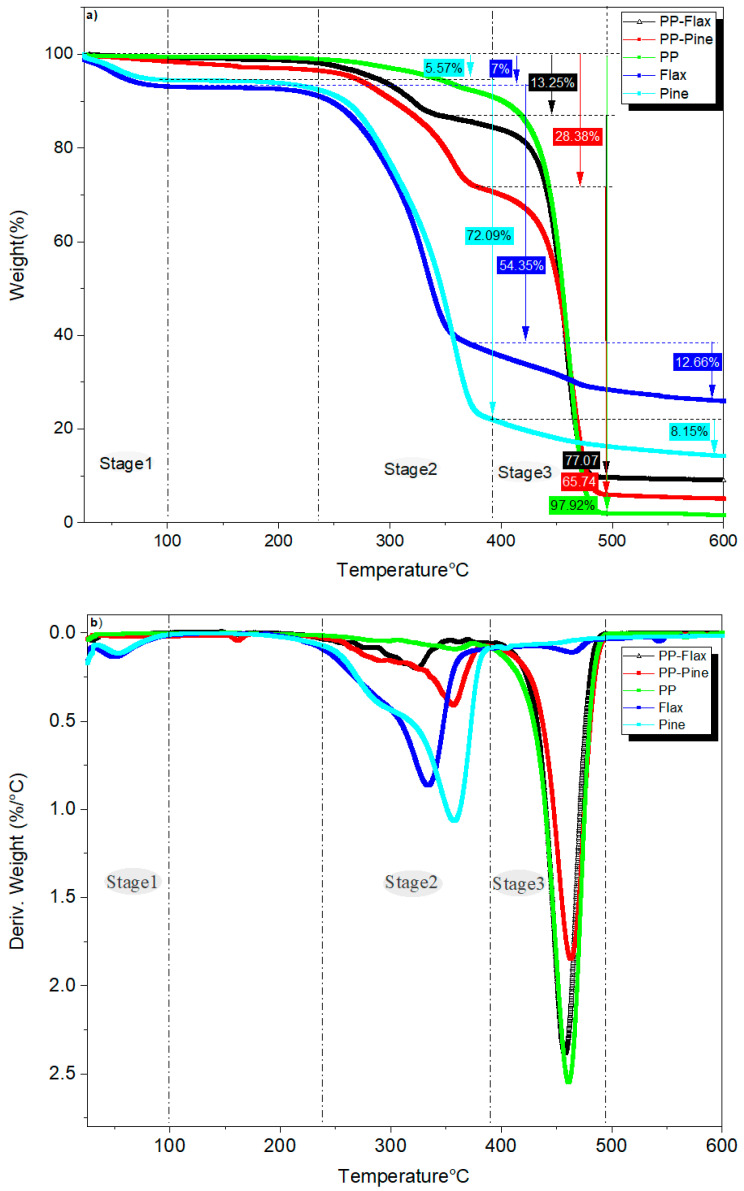
(**a**) Thermogravimetric curves and (**b**) first derivatives for composite pellets and their constituents.

**Figure 6 materials-17-02474-f006:**
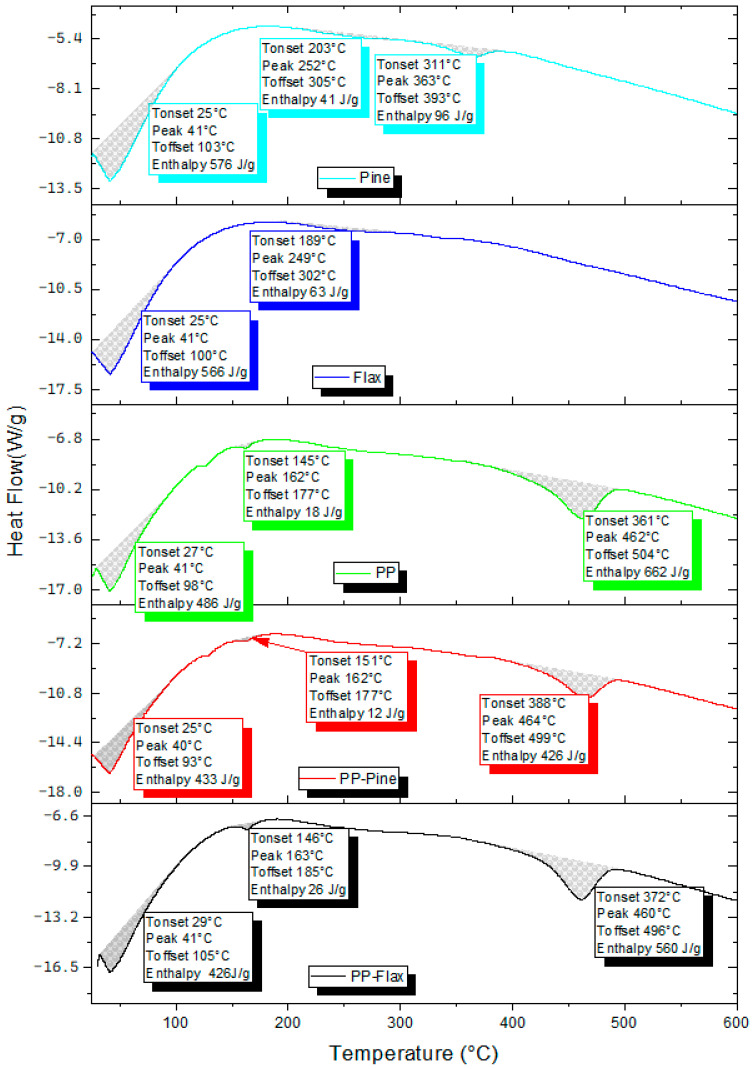
DSC curves for composite pellets and their constituents.

**Figure 7 materials-17-02474-f007:**
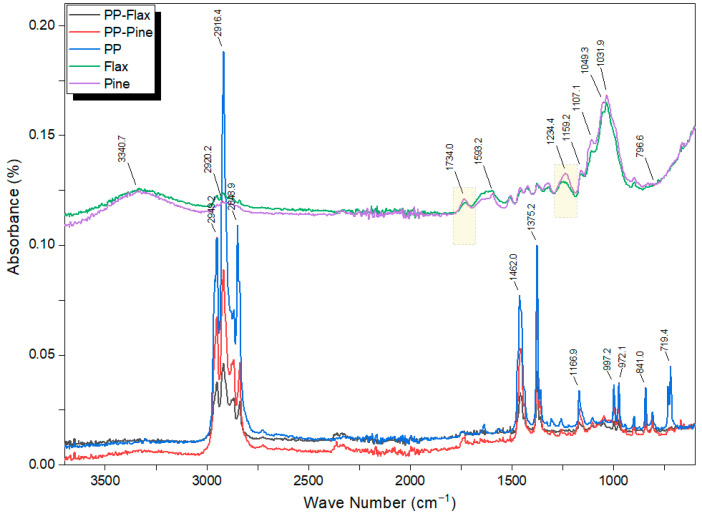
ATR-FTIR spectra for PP–pine and PP–flax composite pellets, along with their individual constituents: pure PP, pine fibres, and flax fibres.

**Figure 8 materials-17-02474-f008:**
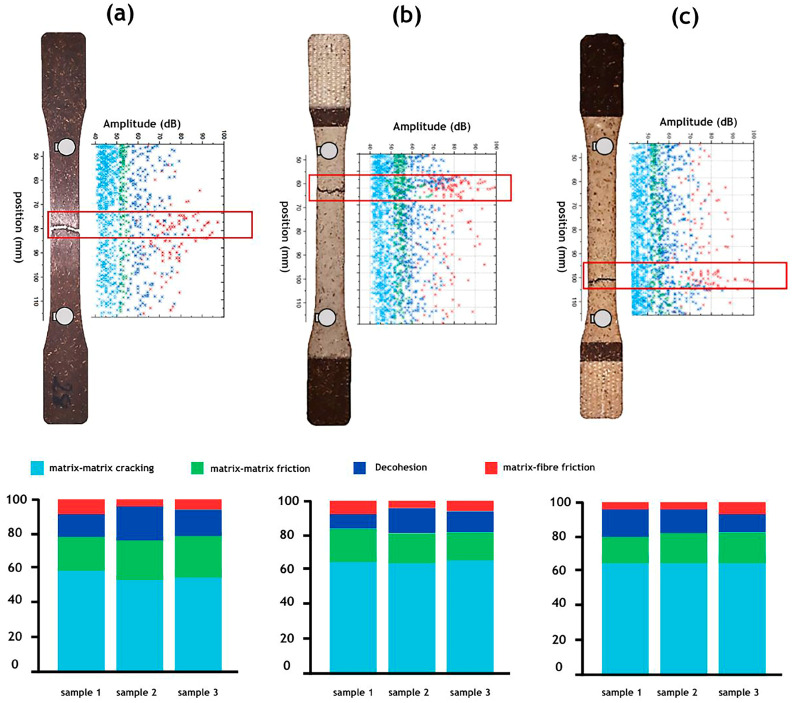
Amplitude versus location of acoustic events recorded during a tensile test on the PP–flax sample for (**a**) the non-aged condition, (**b**) Condition 1, and (**c**) Condition 2 compared with the sample picture after failure.

**Figure 9 materials-17-02474-f009:**
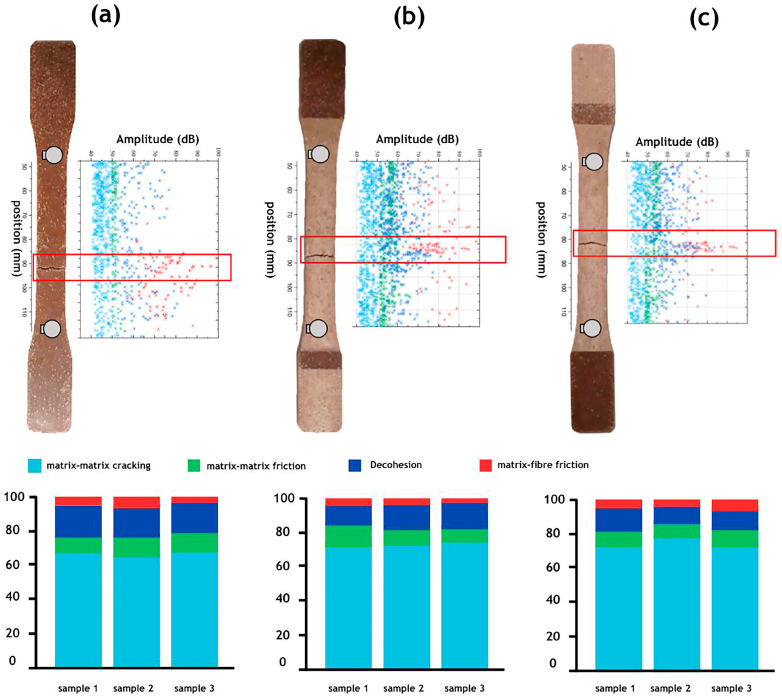
Amplitude versus location of acoustic events recorded during a tensile test on the PP–pine sample for (**a**) the non-aged condition, (**b**) Condition 1, and (**c**) Condition 2 in comparison with the sample picture after failure.

**Table 1 materials-17-02474-t001:** DSC curves for composite pellets and their constituents.

Materials	Temperature (°C)			Type of Peak	Enthalpy (AH) (J/g)	Crystallinity (%)
	Onset	Peak	Offset			
Flax	25.2	41.2	100.5	Endothermic	566.16	-
	189.9	249.1	302.3	Endothermic	63.74	
Pine	25.6	41.5	103.3	Endothermic	576.6	-
	203.2	252.9	305.3	Endothermic	41.47	
	311.2	363.4	393.2	Endothermic	96.84	
PP	27.9	41.1	98.00	Endothermic	486.76	-
	112.7	127.0	137.97	Endothermic	16.68	-
	145.9	162.0	177.97	Endothermic	18.34 (MPE)	9
	361.7	462.1	504.05	Endothermic	662.12	-
PP–flax	29.9	41.8	105.5	Endothermic	426.93	-
	146.4	163.8	185.5	Endothermic	26.22 (MPE)	12
	372.1	460.8	496.6	Endothermic	560.47	-
PP–pine	25.40	40.96	93.36	Endothermic	433.73	-
	117.9	128.3	140.9	Endothermic	12.81	-
	151.2	162.6	177.4	Endothermic	12.10 (MPE)	6
	388.6	464.0	499.4	Endothermic	426.94	-

MPE: Melting Peak Enthalpy.

## Data Availability

Data are contained within the article.

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
