# Peer review of "The Effects of Lignin on the Thermal and Morphological Properties and Damage Mechanisms after UV Irradiation of Polypropylene Biocomposites Reinforced with Flax and Pine Fibres: Acoustic Emission Analysis"

_materials, 2024, doi:10.3390/ma17112474_

Round 1

Reviewer 1 Report

Comments and Suggestions for Authors

This manuscript can be accepted after minor revisions.

1. The abstract needs to be  further summarized.

2. Language needs to be comprehensively revised.

3. The results section needs to require the comparative analysis

Comments on the Quality of English Language

 Minor editing of English language required

Author Response

We are very much thankful to the reviewers for their deep and thorough review. In the light of their useful suggestions and comments we have thoroughly revised our present research paper. We hope our revision has improved the paper quality to a level of their satisfaction. 

Reviewer 2 Report

Comments and Suggestions for Authors

Review Report

In this Research article, “Effects of lignin on the thermal and morphological properties and damages mechanisms after UV irradiation of polypropylene biocomposites reinforced with flax and pine fibers: acoustic emission analysis” The authors examine how lignin content affects the degradation of polypropylene/flax (PP-flax) and polypropylene/pine (PP-pine) fiber composites under UV aging. Pine fibers, with more lignin, showed higher mass loss during thermal decomposition compared to flax fibers. PP-flax exhibited the highest crystallinity. Higher lignin content in PP-pine delayed degradation and reduced microcrack formation. This study sheds light on UV degradation in bio composites and underscores the importance of fiber composition for material performance.

Overall, the paper is well-presented and provides valuable insights. However, before we can proceed with acceptance, there are some corrections and revisions that need to be made to ensure the accuracy, clarity, and rigor of the manuscript.

Top of Form

Introduction:

The introduction should be reconstructed to present a coherent literature review. The last paragraph of the introduction in a research paper typically serves to highlight the gap in the existing literature that the current study aims to address, provide a brief overview of the methodology employed, and outline the specific objectives or hypotheses of the research. Additionally, it may briefly mention the significance of the study and its potential contributions to the field. It may help the authors by answering the following questions:

Why are these works relevant?

Which specific problems were addressed from previous results.?

How are the previous results related to the latest work?

Experiments Top of Form

2.1 Materials, specimens, and tensile testing:

In the materials section, it is imperative for the authors to furnish comprehensive technical details regarding the materials employed in this research endeavor. Given that the materials under investigation are natural fibers, their performance is subject to the influence of numerous factors, including environmental conditions, the method of raw material preparation, and temperature conditions during processing and testing. The environmental conditions encompass factors such as humidity levels, exposure to sunlight, and variations in ambient temperature, all of which can affect the physical and mechanical properties of the fibers. In the materials section the authors must provide the technical information about the materials used in this research work. Since these are natural fibers, there are several factors impact on their performance such as Environmental conditions, preparation of raw materials, temperature conditions etc.

2.2 Characterization methods

1.     In the Characterization Methods section, it is important to address any standards followed for testing methods to ensure consistency, reproducibility, and comparability of results.

  1. The absence of standardized testing procedures can introduce variability and uncertainty into the experimental outcomes.
  2. It would be more appropriate to present SEM-XRD spectroscopic analysis to determine the chemical composition of individual materials.
  3. It is better to avoid general statements in the descriptions; otherwise, the authors must provide the references throughout the manuscript.

3. Results and Discussion

The results are well-presented for comprehension, yet there is a notable absence of certain details, such as the number of samples tested in each case. If multiple samples were tested, it is imperative to include error bars for clarity and accuracy. Some additional pointes to be considered.

1. It is essential to provide a clear explanation of why this particular test was chosen and how it contributes to addressing the research objectives or hypotheses. Understanding the significance of the test in the context of the study helps readers grasp its relevance and importance in advancing knowledge in the field. As it was explained in 3.4 section, in comparison with previous research work.

2. it will be more appropriate if authors provide more detail on the unique characteristics of flax and pine fibers observed in the microscopic images and also What are the implications of the AE test results for the design and engineering of UV-resistant bio composites.

 4. Conclusions

Overall, the research provided a detailed summary of the effect of plant fiber chemical composition on the UV aging behaviour of bio composites reinforced with flax and pine fibers. Here are some specific errors and areas for improvement:

  1. "Following the detailed analysis of PP-flax and PP-pine fibre composites and their components, several significant conclusions can be drawn." - This sentence could be clearer and more direct. It could be revised to something like, "After analyzing PP-flax and PP-pine fiber composites and their components in detail, several significant conclusions emerged."
  2. "By precisely identifying the composition of the granules, we could better understand the thermal behaviour and potential damage mechanisms, which are crucial for the development of resistant materials." - The term "granules" seems out of place here and might need clarification. It's also unclear what "which" refers to in the second part of the sentence.
  3. "TG and DSC analysis confirmed the complexity of the thermal degradation processes of the composites, highlighting the different stages of decomposition of the individual constituents." - This sentence is grammatically correct but could be more specific about what TG and DSC analyses entail and how they contributed to understanding thermal degradation.
  4. "Additionally, ATR–FTIR spectra identified functional groups present in natural fibres and PP, thus strengthening the understanding of the interaction between phases of the composite." - "Additionally" could be replaced with a more specific transition word or phrase. Also, it's not clear what "PP" refers to without prior context. Providing a brief explanation or spelling it out (e.g., polypropylene) would enhance clarity.
  5. "Lastly, the results of the AE tests revealed similar damage mechanisms between the two composites, highlighting their sensitivity to UV-induced microcracks." - "Lastly" could be replaced with a more appropriate transition phrase. Additionally, it would be helpful to specify what "AE tests" stands for and briefly explain their purpose.

To bolster the scientific robustness of the research, the authors should integrate practical application insights before the conclusion section. This involves delving into technical details to illustrate how the findings can be applied in real-world contexts. Additionally, addressing key questions arising from the study's results will enhance its relevance and impact.

1. How do the findings of this study contribute to the development of more durable bio composite materials?

2. How do TG and DSC analyses contribute to understanding the thermal degradation processes of bio composites?

Author Response

(The authors gave the same response as above.)

Reviewer 3 Report

Comments and Suggestions for Authors

The researchers have submitted a noteworthy manuscript that delves into a crucial topic- using flax and pine fibres to reinforce polypropylene biocomposites, a method that could significantly reduce the degradation and appearance of microcracks. However, the manuscript requires some revisions before it can be considered for publication.

1. Please revise the whole manuscript by avoiding "we" or the first person singular or plural.

2. While providing information about the materials and properties, the abstract could benefit from a more explicit statement about the novelty of the current study and its necessity. Please consider adding one or two lines that briefly explain the unique contribution of the current study. Revising Lines 37-40 might be highlighted in the abstract. 

3. Please revise the paragraph from lines 73 to 90, as it is the core of the manuscript and the reason for conducting the entire analysis. The presentation of the paragraph has been insufficient, and it would be better to avoid using "we" and instead use phrases like "In this study" or "the current study". Additionally, it was suggested to use words like "according to the literature review" or "studies conducted thus far" to support "our beliefs". Please make sure to correct any spelling, grammar, and punctuation errors. 

4. Please remove lines 91 to 95. They were not required in the manuscripts unless they were book chapters.

5. Please make some revisions to the text that mentions Figure 1. The current text only mentions shaping the tensile specimens, but Figure 1 includes two parts, namely Figure 1(a) and Figure 1(b). Please add information about both parts in the main text. Also, the images in Figure 1 were unclear; please revise them.

6. Line 111, please add the sieve size used in the process. 

 7. The injection moulding was mentioned only in temperature and time. Please add the pressure and other relevant details like cooling and waiting time after moulding before the sample was removed from the mould. (118- 119).

8. True density measurement: It is recommended that line 123 be revised to incorporate all types of powders as provided in Figure 4, which includes pure PP, flax, pine, and composites. Additionally, the number of samples utilised for each powder should be included, and the statistical analysis with an error bar should be added. Given that the product is commercial, this information would provide ample data. Also, in section 3.2, please discuss other findings on how noteworthy the current product is compared with other composites or similar studies on PP-flax and -pine (Line 194 - 200) with citations. 

9. Thermal assessment:

9.1 Please add the full abbreviation of DSC (Line 126). However, with DSC, it would be great to include the heating and cooling cycles rather than one heating cycle. That will give more precise data about the composites. 

9.2 Please check the font of line 128. 

9.3 Line 218: Please add the wt% of char or black colour content found after TGA analysis. 

9.4: Line 226: Please revise the units because it is supposed to be in C, not in K

9.5: Line 227: Please check which word is suitable. Is it woody or wood biomass? 

9.6 Line 233: the case of hydrogels will be different from the solid-state materials with less water content. Therefore, please revise the lines and add relevant recent studies that may be based on recycled products.

10: Section 3.5: 

10.1 Please put the calculated enthalpy and crystallinity data in a table rather than a statement (Line 245, 252). 

10.2: Although the section was talked about with appropriate citations and comparisons with literature reports, please look at section 3.2 carefully. According to the results, the readability is low. To improve the section, it is recommended to focus on comparing pure PP with composites and flax and pine with composites rather than mixing them all into a single paragraph. Additionally, presenting the calculated data (Table 1 with data on crystallinity) in a table format would enhance the readability and allow the use of only the relevant data points in the discussions. Table 1: Please delete the data for Ref 19. Use them to discuss the results and compare them to currently obtained data. 

11. FTIR analysis: 

11.1 Please delete - "This method is a robust tool...."

11.2 Line 294: Please recheck that Fig 6 was relevant to FTIR. The manuscript shows data about acoustic analysis. 

11.3 Please check the units and be consistent is it J/g oe J.g-1, as the lines from 299 to 303 shows cm-1

11.4 Section 3.6 adequately explains the FTIR data obtained. However, please also discuss and compare the results with those previously published. 

12. Morphological analysis

Please add a table based on analysing three or more data points from the micrograph and include the L/D ratio using the standard deviation approach. Also, please revise the images. The micrographs could be more precise, including figures (a) and (b) highlighting what fibre it is, pine or flax. If authors can access SEM/FESEM, please incorporate the micrographs obtained from SEM. SEM software might have an easy way to calculate the data, or with optical micrographs, ImageJ software (free version available) can obtain n=3 data points rather than single fibres. 

13. Artificial weathering: 

Please add the results from artificial weathering in a separate section instead of incorporating them with acoustic emission. 

13.1 Please add the weight decrease or degradability levels through exposure to the varied levels of the weathering. 

13.2 Please include a discussion with relevant support. Lines 346 - 358 were stated in a generalised format without any technical aspects to the current study. 

14. Acoustic emission

14.1 PP30-F and PP30-P, please include what they are in the material sections. Both the samples were brought into the context without appropriate introduction. This must be clarified for non-technical readers and grad students who want to follow the author's papers to test materials using acoustic emissions. 

14.2 Figures 6 and 7 needed clarity, particularly with amplitude graphs. 

14.3. Please add a discussion about the results in section 3.7. 

15. Conclusions 

The results were adequately summarised; however, further study should be done after lines 386 - 389. In addition, please explain how the current is analysed to find the optimum environmental conditions and other related findings in one or two lines in the conclusions. 

16: Please be consistent with the graph colour and increase the DPI to provide clear and concise images. In addition, sections 2.1 to 2, materials and methodology, should be revised. Therefore, authors can specify all the 2.1 materials and variations of the materials used in the testing. Methodology as 2.2. 

17: As the authors produced tensile specimens it would be great if authors could do tensile testing using UTM and incorporate the results. Furthermore, that might be helpful to compare with acoustic emission comparison cracking.  The tensile test results can be integrated without the comparison as well. 

18. if the authors can access the Melt Flow Index, please incorporate the as-obtained biocomposite sample and separated PP results.

Comments on the Quality of English Language

Please carefully check the grammatical errors.  

Author Response

(The authors gave the same response as above.)

Reviewer 4 Report

Comments and Suggestions for Authors

The article discusses the effects of lignin on the thermal and morphological properties as well as damage mechanisms after UV irradiation of polypropylene biocomposites reinforced with flax and pine fibers. Here are some questions ad comments:

1.       What is the main focus of the study conducted in the article should be written in the abstract.

2.       Line 41: the reference is missing.

3.       Line 48: the reference is missing.

4.       The aim should be written at the end of the Introduction part.

5.       Line 91 – 95 is not necessary.

6.       How does the lignin content affect the thermal degradation of the polypropylene/flax and polypropylene/pine fiber composites?

7.       What are the differences in thermal decomposition mass loss between pine and flax fibers, and what is attributed to these differences?

8.       How does the study demonstrate the influence of lignin content on degradation and microcrack formation in the biocomposites?

9.       What role does acoustic emission analysis play in understanding the degradation process of the biocomposites? It should be clearly stated in the article.

These questions cover various aspects of the article, including its methodology, findings, and implications. They can help facilitate a deeper understanding of the research presented.

Author Response

(The authors gave the same response as above.)

Reviewer 5 Report

Comments and Suggestions for Authors

The manuscript by Toubal describes the effects of natural fibers (not only of lignin) on  the properties  and damages after UV irradiation of PP doped with flax and pine fibres. I have several concerns for which I cannot recommend publication in Materials: the manuscript lacks a figure, details and literature comaprison.

The drying procedure of the isolated components is not reported. This may affect the all the test and hamper comparison. Moreover, the use of solvents as xilene can alter the composition of the natural fibers, making less meanigful the TGA observation (e.g. the first weight loss step) . 

 The sentence “Exposure times in most weathering cycles, except for the dew cycle and fluorescent UV-condensation type (ASTM G53) exposure, range from 1000 to 2000 h. We conducted tests with exposures of up to 1400 hours, adhering to the standard practices outlined in the literature.“ is not clear. Why "except"? How many hours where used? Up to 2000h or 1400h?

How can “A precise balance of L/D proves essential to optimizing the mechanical performance of composites.”? Would it be possible to precisely tuning  dimensions of fibers without changing their nature, so to optimize the properties?

Density of the composites can be different also because of differences in fibers’ density. However, this is not mentioned as a possibility, why

The difference in density are small and comparable to PP despite the relatively high content of fibers. Can the authors comment on this? How much is the error on the measurements?

A remark on the fastest degradation step in the TGA: the behaviour is very similar for the two fibers. The different lignin content has only a minor effect. 

I don’t understand “a 72%  mass loss indicates a delay in decomposition compared to flax fibre” : why a delay? 72% vs 54% suggest that it is more degraded, so it should be quicker (see also below)

20°k? Is it 20°C?

“Slower thermal degradation is observed for PP-pine pellets, resulting in a 28% mass loss, while PP-flax pellets lose 13% mass.: “I would say it is faster in all range. Please clarify.

Lignine should left more reside than cellulosic materials (as stated later in the text “lignin is the most thermally stable component of natural fibres”. In Ref 22, the most stable hydrogels, indeed, have higher content of residue. Why, in the present case, PP-pine has a lower residue? If, as stated later in the text, crystallinity can have a role, so I think the discussion on thermal stability should have been done considering all thermal data, to avoid misunderstanding

PAM and CMX acronyms should be explained

DSC is used to discuss water content. How can water content be still considered those of natural fibers even after xylene treatment (and drying, if any) ?

Figure of FTIR (should be Figure 6) is missing. So, I cannot evaluate the big number of peaks which have been reported. And, more importantly, it not possible to evaluate the final remark “Additionally, ATR–FTIR spectra identified functional groups present in natural fibres and PP, thus strengthening the understanding of the interaction between  phases of the composite.”

Paragraph 3.67 should introduce and describe the type of test performed.

No comparison with PP and with other systems reported in the literature has been included. No comment on the obtained values has been reported. It is not possible to fully evaluate the quality of the materials and if it displays appealing characteristics. In this scenario, also the term “reinforced”, used throughout the text might be a speculation and should be carefully evaluated after comparison with PP.

Comments on the Quality of English Language

English grammar should be slightly revised

Author Response

(The authors gave the same response as above.)

Round 2

Reviewer 2 Report

Comments and Suggestions for Authors

The paper was improved significantly therefor I am recommending for publication.

Author Response

We are very much thankful to the reviewers for their deep and thorough review

Reviewer 3 Report

Comments and Suggestions for Authors

The authors revised the manuscript based on feedback, improving its technicality and readability. It can now be accepted for publication.

Comments on the Quality of English Language

N/A

Author Response

(The authors gave the same response as above.)

Reviewer 5 Report

Comments and Suggestions for Authors

The manuscript has been improved. I appreciate the efforts made by the authors. 

However, I recommend to add a note on DSC results and on water content as a result of post-processing).

I also suggest to add a final discussion of the overall outcome, i.e. on the impact of the fibers, not only with respect to previous research work by the authors, but also with respect to different PP-based composites. See, for example: doi: 10.3390/polym13223931.

Comments on the Quality of English Language

Unclear sentences are now improved.

Author Response

 We are very much thankful to the reviewers for their deep and thorough review. In the light of their useful suggestions and comments we have thoroughly revised our present research paper. We hope our revision has improved the paper quality to a level of their satisfaction.  Number wise answers to their specific comments/suggestions/queries are as follows.
